# Diffusion Domain Teacher: Diffusion Guided Domain Adaptive Object Detector

Boyong He*
Xiamen University
Institute of Artifcial
Intelligence
Xiamen, China
boyonghe@stu.xmu.edu.cn

Yuxiang Ji*
Xiamen University
Institute of Artifcial
Intelligence
Xiamen, China
yuxiangji@stu.xmu.edu.cn

Zhuoyue Tan
Xiamen University
Institute of Artifcial
Intelligence
Xiamen, China
tanzhuoyue@stu.xmu.edu.cn

Liaoni Wu†
Xiamen University
Institute of Artifcial
Intelligence
School of Aerospace
Engineering
Xiamen, China
wuliaoni@xmu.edu.cn

## Abstract

Object detectors often suffer a decrease in performance due to the large domain gap between the training data (source domain) and real-world data (target domain). Diffusion-based generative models have shown remarkable abilities in generating high-quality and diverse images, suggesting their potential for extracting valuable features from various domains. To effectively leverage the cross-domain feature representation of diffusion models, in this paper, we train a detector with frozen-weight diffusion model on the source domain, then employ it as a teacher model to generate pseudo labels on the unlabeled target domain, which are used to guide the supervised learning of the student model on the target domain. We refer to this approach as *Diffusion Domain Teacher* (**DDT**). By employing this straightforward yet potent framework, we significantly improve cross-domain object detection performance without compromising the inference speed. Our method achieves an average mAP improvement of 21.2% compared to the baseline on 6 datasets from three common cross-domain detection benchmarks (*Cross-Camera, Syn2Real, Real2Artistic*), surpassing the current state-of-the-art (SOTA) methods by an average of 5.7% mAP. Furthermore, extensive experiments demonstrate that our method consistently brings improvements even in more powerful and complex models, highlighting broadly applicable and effective domain adaptation capability of our DDT.

## CCS Concepts

• **Computing methodologies → Image representations**; **Object detection**.

## Keywords

Unsupervised domain adaptation; Object detection; Diffusion model

---

*Contribute equally to the work.
†Corresponding author.

---

**ACM Reference Format:**
Boyong He, Yuxiang Ji, Zhuoyue Tan, and Liaoni Wu. 2024. Diffusion Domain Teacher: Diffusion Guided Domain Adaptive Object Detector. In *Proceedings of the 32nd ACM International Conference on Multimedia (MM '24), October 28-November 1, 2024, Melbourne, VIC, Australia.* ACM, New York, NY, USA, 10 pages. https://doi.org/10.1145/3664647.3680962

## 1 Introduction

Object detection is a fundamental task in computer vision, with its applications permeating an array of real-world scenarios. There have been impressive strides and significant achievements in object detection, leveraging both Convolutional Neural Networks (CNNs) [18, 42, 53, 65] and transformer-based models [4, 85]. Nonetheless, these data-driven detection algorithms wrestle with the challenging issue of domain shift: the large gap between the training data (source domain) and the testing environments (target domain) frequently results in a substantial decline in detection accuracy. This obstacle is ubiquitous across various sectors, including robotics, autonomous driving, and healthcare, and it poses a formidable barrier to the widespread applications of object detection in practice. Consequently, the deployment of domain adaptation techniques, aimed at minimizing domain disparities, has become essential to boost the robustness and generalizability of models across diverse environments.

Unsupervised Domain Adaptation (UDA) methodologies have surged to the forefront of research, taking advantage of the sparse labeled data from the source domain in conjunction with copious unlabeled data from the target domain to significantly enhance cross-domain detection performance. Current UDA tactics have explored a variety of strategies including domain classifiers [10, 25, 59, 81, 84], graph matching [36, 38, 39], domain randomization [32], image-to-image translation [26], and self-training frameworks [6, 14, 41, 56]. These techniques have been crucial in achieving notable advancements in cross-domain object detection.

Moreover, diffusion-based generative models [24, 54, 61] have showcased remarkable capabilities in generating high-quality and diverse images, signaling their vast potential for a spectrum of downstream applications. Some works [1, 66, 71] have already harnessed diffusion models for a breadth of tasks. This evidence suggests a promising avenue for employing these models to bolster cross-domain detection efficacy. Nevertheless, the step-by-step inference process of these models is not fast enough to meet the immediate processing needs of object detection. Although there

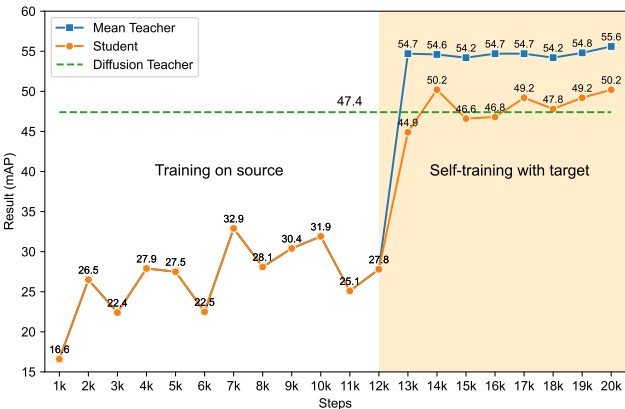

**Figure 1: Evaluation results on Clipart [28] during training.** It is evident that the performance of the **student** significantly improves after entering self-training, even surpassing the **diffusion teacher**, and the **mean teacher** exhibits better performance compared to the student.

has been some effort to adapt diffusion models for image generation and manipulation, as seen with tools like LoRA [27] and ControlNet [76], there is a lack of research on applying diffusion models to cross-domain detection.

Fortunately, current UDA methods provide us with valuable insights. Specifically, we draw inspiration from previous state-of-the-art (SOTA) approaches [3, 14, 41] and adopt the Mean Teacher [64] self-training framework, where the teacher model generates pseudo labels for the supervised learning of student model on the target domain. The weights of the teacher model are typically updated through Exponential Moving Average (EMA) by the student model. This consistency-based self-training approach allows the student model to progressively learn from the target domain, thereby improving the performance of the detector in cross-domain detection.

In our approach, we freeze all parameters of the diffusion model and extract intermediate feature from the upsampling structure of the U-Net [55] architecture during the inversion process. These features are then passed through a bottle-neck structure to generate hierarchical features similar to a general backbone for downstream detection tasks. This enables effective training and fine-tuning of the diffusion model with a small number of parameters, and yields discriminative feature for classification and regression tasks, leading to improved performance in cross-domain detection. The mean teacher, updated through EMA from the student model, further enhances stability and generalization.

Through the detector with the diffusion backbone for feature extraction struggles to match or surpass the performance of general backbones like ResNet [22] on intra-domain. However, in the target domain, the performance of the diffusion detector surpasses them greatly. This strongly confirms the diffusion model is an incredibly powerful and highly generalized feature extractor. Furthermore, it is even more remarkable that the diffusion teacher model continues to enhance the cross-domain performance of stronger backbones, all without any increase in additional inference speed.

The contributions of this paper can be summarized as follows:

- We introduce a frozen-weight diffusion model as backbone, which efficiently extracts highly generalized and discriminative feature for cross-domain object detection. Notably, the diffusion-based detector, trained exclusively on the source domain, demonstrates exceptional performance when applied to the target domain.
- We incorporate the diffusion detector as a teacher model within the self-training framework, providing valuable guidance supervised learning of the student model on the target domain. This integration effectively enhances cross-domain detection performance without any increasing of inference time.
- We achieve substantial improvements in cross-domain detection. Our method achieves an average mAP improvement of 21.2% compared with the baseline, and surpassing the current SOTA methods by 5.7% mAP. Further experiments demonstrate that the diffusion domain teacher consistently enhances cross-domain performance for detectors with stronger backbones, leading to superior results in the target domain.

## 2 Related Work

### 2.1 Object Detection

Object detection aims to locate and classify objects in given images. Deep convolutional neural networks [22, 60] have revolutionized this teak and is widely applied in real-world applications. Faster R-CNN [18], a prominent two-stage detection method, employs a region proposal network (RPN) to generate candidate regions, followed by region of interest (ROI) refinement to determine the final bounding boxes and classes. Some research is focused on improving the precision and efficiency of two-stage methods [2, 51]. In addition, researchers are investigating single-stage detectors [42, 53, 67] aimed at simplifying the detection process by integrating box regression and classification. Moreover, anchor-free detectors [65, 77] that eliminate the reliance on predefined anchors have garnered significant attention. Recent research trends involve the adoption of transformer-based end-to-end detectors [4, 75, 85], which reconceptualize the detection task as a set prediction problem, thereby obviating the necessity for traditional handcrafted components such as anchor generation and non-maximum suppression (NMS).

### 2.2 Domain adaptation Detection

Although object detection has made significant advancements, performance can suffer greatly due to domain shifts between training and test data. To address this issue, UDA aims to mitigate the impact of domain shifts by leveraging labeled source data and unlabeled target data. Initially, studies inspired by GANs [20] introduced domain adversarial training [17], which minimized domain gaps by extracting invariant feature. This approach is later adapted to detection tasks [10, 25, 59, 81, 84], resulting in notable improvements in performance on the target domain. Some methods [26] focus on reducing inter-domain differences at image level, by applying image-to-image translate like CycleGAN [83].

Recently, self-training domain adaptation methods [3, 13, 14, 41] have achieved better results in cross-domain object detection by optimizing the learning of pseudo labels in the target domain. For

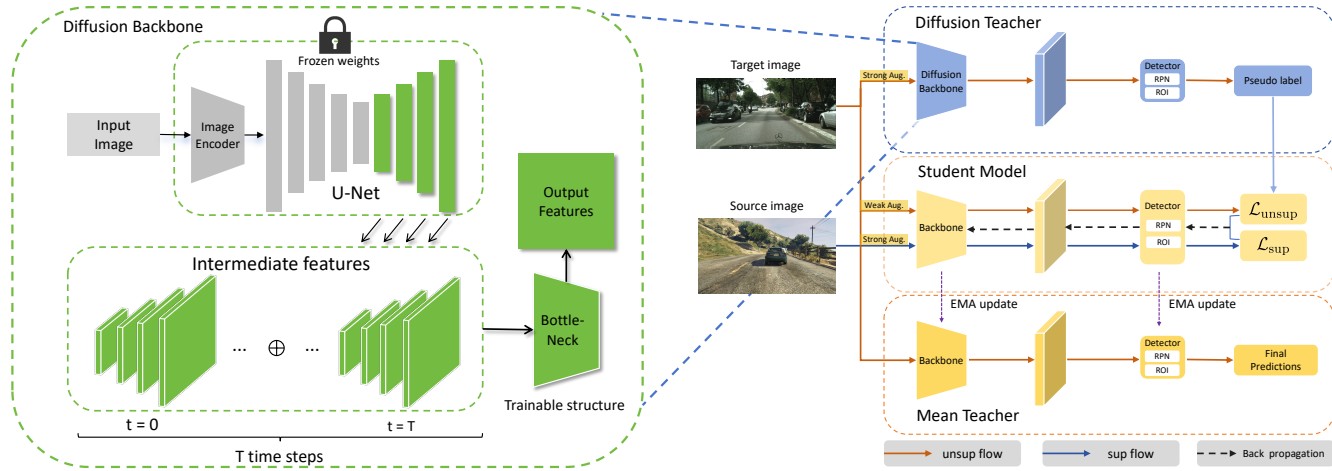

**Figure 2: Overview of our proposed Diffusion Domain Teacher (DDT). Left:** We employ a frozen-weight diffusion model with bottleneck as the **diffusion backbone**, which acquires and aggregates intermediate feature from the U-Net [55] during the inversion process at $T$ time steps for detection. **Right:** We use the **diffusion teacher**, which is a detector with the diffusion backbone, and applied it in self-training to generate pseudo labels for unlabeled target, guiding the learning of the student. By EMA updated from the student model, **mean teacher** is refined and serves as the final model, resulting in improved cross-domain detection results.

example, AT [41] combines domain adversarial training and self-training to improve the quality of pseudo labels. UMT [13] utilizes consistency learning for a teacher model to generate high-quality pseudo labels to improve cross-domain detection. CMT [3] introduces contrastive learning to optimize the utilization and alignment of features from the source and target domain. HT [14] optimizes the generation of pseudo labels by applying consistency measures in regression and classification. Overall, self-training methods in cross-domain object detection enhance the detection performance in the target domain by improving the quality of pseudo labels, with the teacher model being updated from the student model.

### 2.3 Diffusion Models

Diffusion models [24, 52, 54, 58, 61] have achieved impressive results in image generation, surpassing previous models like GAN [20]. With their strong generative and generalization capabilities, some research has begun to explore the potential of diffusion models in feature representation and their application to downstream tasks. For instance, DDPMSeg [1] and ODISE [71] utilize feature extracted from diffusion models for semantic and panoptic segmentation tasks, respectively. DIFT [63] and HyperFeature [49] use the diffusion model to discover correspondences in images. This inspires us to consider the application of diffusion models for improving cross-domain detection tasks.

### 3 Approach

In this section, we present our Diffusion Domain Teacher (DDT) framework in detail. First, in Sec. 3.1, we review the formulation of Unsupervised Domain Adaptation Detection (UDAD). Then, in Sec. 3.2, we provide a detailed description of how the frozen-weight diffusion model serves as a feature extractor, producing hierarchical features, to adapt to the detection task. Furthermore, in Sec. 3.3,

we explain the application of the diffusion teacher detector in thr self-training framework, where pseudo labels generated on the unlabeled target domain guide the supervised learning of the student model. Finally, we summarize the total training objective.

### 3.1 Formulation of Unsupervised Domain adaptation Detection

To be specific, we denote a given set of $N_s$ samples $\mathcal{S} = \left\{ X_s^i, Y_s^i \right\}_{i=1}^{N_s}$ as source domian, where $X_s^i$ represents an image and $Y_s^i$ represents the bounding box with category labels in the respective image. Similarly, we denote the target domain data as $\mathcal{T} = \left\{ X_t^i \right\}_{i=1}^{N_s}$, which consists of $N_t$ unlabeled samples. Exactly, the distribution of $\mathcal{S}$ and $\mathcal{T}$, including the distributions of images from $P(X_s)$ and $P(X_t)$ (e.g., style, scene, weather), labels $P(Y_s)$ and $P(Y_t)$ (e.g., the shapes, sizes, and density of instance), and even the scales of $N_s$ and $N_t$ are different, denoted as $P(\mathcal{S}) \neq P(\mathcal{T})$, is what we refer to as a cross-domain detection problem. Furthermore, relying solely on supervised learning from the labeled source domain results in an inherent bias towards source domain in cross-domain detection. Domain adaptation for detection aims to improving the performance on the target domain by reducing the dissimilarity between $\mathcal{S}$ and $\mathcal{T}$, seeking to a domain-invariant detector.

### 3.2 Fozen-Diffusion Feature Extractor

Diffusion generative models [24, 54, 61] aim to minimize the discrepancy between the distribution of images generated by the model, denoted as $P_\theta(x)$, and the distribution of the training data, denoted as $P_{\text{data}}(x)$. During training, gaussian noise of varying magnitudes is added to the clean training data, commonly referred to as *diffusion* process. The diffusion process starts with a clean image $x_0$ from the training data and generates a noisy image $x_t$ by mixing

$x_0$ with noise of different magnitudes:

$$x_t = \sqrt{\bar{\alpha}_t}x_0 + \sqrt{1-\bar{\alpha}_t}\epsilon \tag{1}$$

where $\epsilon \sim \mathcal{N}(0, \mathbf{I})$ represents randomly sampled noise, and $t \in [0, T]$ denotes the time step, where larger values correspond to adding more noise. The amount of noise added is determined by $\alpha_t$, which is a predefined noise schedule, and $\bar{\alpha}_t = \alpha_1\alpha_2\ldots\alpha_t$. The model $f_\theta$ is trained to predict the input noise $\epsilon$, given $x_t$ and $t$, typically using structures like U-Net [55].

The iterative process of the diffusion model poses challenges when directly applied to downstream supervised tasks. We extract intermediate feature at a specific time step $t$ during the inversion process, and apply these features for regression and classification tasks in the detection task. Specifically, we append a input noise corresponding the time step $t$ to the input image, shift it to $x_t$ and then input it along with $t$ into $f_\theta$ to extract activation layers as intermediate feature. More specifically, we apply the intermediate feature from the four stages of the upsampling process in the denoise network U-Net [55]. For each input image, we concatenate multiple time step feature together and employ a bottle-neck structure to project the feature into hierarchical layers with a channel size of [256, 512, 1024, 2048], similar to the output of ResNet [22], which is directly applied to the object detection task, as shown in the left side of Fig. 2.

## 3.3 Diffusion Teacher Guided Self-training Framework

We employ a detector that extracts feature using the diffusion model and trained on the source domain as the teacher model, denoted as $\mathcal{F}_{\text{diff}}$. It is used to generate pseudo labels $\bar{Y}_t$ on the target domain $\mathcal{T}$, where $\bar{Y}_t = \mathcal{F}_{diff}(X_t)$. These pseudo labels are constructed to form a new dataset $\bar{\mathcal{T}} = \{X_t^i, \bar{Y}_t^i\}_{i=1}^{N_t}$. Subsequently, we optimize the student model using the pseudo labels. We introduce a hyperparameter $\sigma$ as a threshold for the confidence scores of the output for the teacher model, enabling us to select more reliable pseudo labels.

We define the supervised learning of the student model $\mathcal{F}_{\text{stu}}$ on the source domain as follows:

$$\begin{aligned}\mathcal{L}_{\text{sup}}(X_s, Y_s) =& \mathcal{L}_{\text{cls}}^{\text{RPN}}(X_s, Y_s) + \mathcal{L}_{\text{reg}}^{\text{RPN}}(X_s, Y_s) \\ &+ \mathcal{L}_{\text{cls}}^{\text{ROI}}(X_s, Y_s) + \mathcal{L}_{\text{reg}}^{\text{ROI}}(X_s, Y_s)\end{aligned} \tag{2}$$

where RPN is used to generate potential candidate regions, and ROI performs classification and bounding box regression on these candidate regions to obtain more accurate class and bounding box predictions, denoted as cls and reg, respectively. Similarly, we define the learning of the student model in the target domain as follows:

$$\begin{aligned}\mathcal{L}_{\text{unsup}}(X_t, \bar{Y}_t) =& \mathcal{L}_{\text{cls}}^{\text{RPN}}(X_t, \bar{Y}_t) + \mathcal{L}_{\text{reg}}^{\text{RPN}}(X_t, \bar{Y}_t) \\ &+ \mathcal{L}_{\text{cls}}^{\text{ROI}}(X_t, \bar{Y}_t) + \mathcal{L}_{\text{reg}}^{\text{ROI}}(X_t, \bar{Y}_t)\end{aligned} \tag{3}$$

Then, we employ EMA to update a mean teacher model $\mathcal{F}_{mean}$ by copying the weights from the student model. We define this process as follows:

$$\theta_t \leftarrow \alpha\theta_t + (1-\alpha)\theta_s \tag{4}$$

where $t$ and $s$ represent the parameters of $\mathcal{F}_{mean}$ and $\mathcal{F}_{\text{stu}}$, respectively. By employing EMA to update the mean teacher model,

we aim to create a more stable and robust model by gradually incorporating the knowledge learned by the student model over time. We select the output of $\mathcal{F}_{\text{mean}}$ as result for predicting.

We apply a hyper parameter $\lambda$ to adjust the weights between $\mathcal{L}_{\text{unsup}}$ and $\mathcal{L}_{\text{sup}}$. The final formulation of our comprehensive loss function is summarized as follows:

$$\mathcal{L} = \mathcal{L}_{\text{sup}} + \lambda \cdot \mathcal{L}_{\text{unsup}} \tag{5}$$

In our DDT framework, following [41], we employ *Weak Augmentation* to provide target domain images to the diffusion teacher model for generating reliable and accurate pseudo labels. Simultaneously, we apply *Strong Augmentation* to the images as inputs to the student model, as illustrated in Fig. 2. Specifically, *Weak Augmentation* includes random crop and random horizontal flip, while *Strong Augmentation* involves color transformations such as color space conversion, contrast adjustment, equalization, sharpness enhancement, and posterization, as well as spatial transformations such as rotation, shear, and translation of the position.

## 4 Experiments

### 4.1 Datasets

**Cityscapes.** Cityscapes [12] dataset provides a diverse of urban scenes from 50 cities. It includes 2,975 training images and 500 validation images with detailed annotations. The dataset covers 8 detection categories, using bounding boxes sourced from instance segmentation.

**BDD100K.** BDD100K [73] dataset is a comprehensive collection of 100,000 images specifically designed for autonomous driving applications. The dataset offers detailed detection annotations with 10 categories.

**Sim10K.** Sim10k [30] is a synthetic dataset comprising 10,000 rendered images simulated within the Grand Theft Auto gaming engine, specifically designed to facilitate the training and evaluation of object detection algorithms in autonomous driving systems.

**VOC.** VOC [16] is a general-purpose object detection dataset that includes bounding box and class annotations for common objects across 20 categories from the real world. Following [41], we combined the PASCAL VOC 2007 and 2012 editions, resulting in a total of 16,551 images.

**Clipart.** Clipart [28] dataset comprises 1,000 clipart images across the same 20 categories as the VOC dataset, exhibiting significant differences from real-world images. Following [41], we utilize 500 images each for training and testing purposes.

**Comic.** Comic [28] dataset consists of 2,000 comic-style images, featuring 6 categories shared with the VOC dataset. Following [29], we allocate 1,000 images each for training and testing.

**Watercolor.** Watercolor [28] dataset contains 2,000 images in a watercolor painting style, with 6 categories shared with the VOC dataset. Following [41], we use 1,000 images for both training and testing.

### 4.2 Cross-domian Detection Settings

**Cross-Camera.** We train on Cityscapes [12] (source domain) and validate on BDD100K [73] (target domain) to evaluate the cross-camera detection performance in diverse weather and scene conditions. We focus on the 7 same categories as SWDA [59].

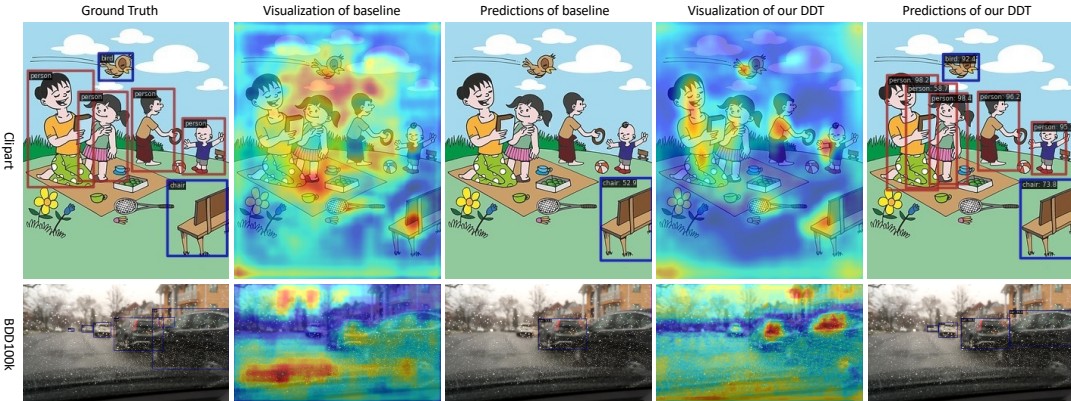

**Figure 3: Qualitative prediction results and feature visualization of baseline and our DDT.** Compared to the baseline, our method focuses more on specific classes of objects in the target domain images, effectively reducing the numbers of false negative on Clipart [12] (**first row**) and BDD100K [73] (**second row**).

**Synthetic to Real (Syn2Real).** We train on Sim10K (source domain) and validate on Cityscapes [12] and BDD100K [73] (target domian)to validate the performance of synthetic-to-real detection. Following SWDA [59], we focus on the shared category *car*.

**Real to Artistic.** We train on the VOC [16] (source domain) and perform validation on the Clipart [28], Comic [28], and Watercolor [28] (target domains) to assess cross-domain detection performance from real-world images to artistic styles. Referring to AT [41] and D-ADAPT [29], we respectively apply the 20, 6, and 6 shared categories between VOC and each of the Clipart, Comic, and Watercolor.

### 4.3 Implementation Details

Following [29, 41, 59], we use Faster R-CNN [18] as the default detector with a ResNet101 [22] backbone pretrained on ImageNet [57], implemented with MMDetection [8]. The training and testing sizes of images are set to (1333, 800) for Cityscapes, BDD100K, and Sim10K, and (1200, 600) for VOC, Clipart, Comic, and Watercolor. The models are trained with 20,000 steps on two 3090 GPUs, with a total batch size of 16. We employ the SGD optimizer with an initial learning rate of 0.02, following the default settings in MMDetection.

In self-training, we refer to the settings in [14, 41] to apply both weak and strong augmentation on the unlabeled target domain. We employ the EMA update parameter $\alpha$ of 0.999 for the mean teacher model updates and simply set the loss weight $\lambda$ to 1. We train exclusively on the source domain for the first 12000 steps and then perform joint training on both the source and target domain for the remaining 8000 steps.

For evaluation, we report the Average Precision (AP) for each object category and the mean Average Precision (mAP) across all categories, with applying an Intersection over Union (IoU) threshold of 0.5.

### 4.4 Results and Comparisons

In this section, we present the evaluation result of our DDT framework along with other SOTA approaches. Current cross-domain object detection methods employ different detectors [18, 45, 65, 85],

**Table 1: Quantitative results on adaptation from** Cityscapes to BDD100K (Cs→B). **The bold indicates the best results.**

| Method | Reference | Detector | bicycle | bus | car | mcycle | person | rider | truck | mAP |
|---|---|---|---|---|---|---|---|---|---|---|
| **DA-Faster** [10] | *CVPR'18* | FRCNN-V16 | 22.4 | 18.0 | 44.2 | 14.2 | 28.9 | 27.4 | 19.1 | 24.9 |
| **SWDA** [59] | *CVPR'19* | FRCNN-V16 | 23.1 | 20.7 | 44.8 | 15.2 | 29.5 | 29.9 | 20.2 | 26.2 |
| **SCDA** [84] | *CVPR'19* | FRCNN-V16 | 23.2 | 19.6 | 44.4 | 14.8 | 29.3 | 29.2 | 20.3 | 25.8 |
| **CRDA** [70] | *CVPR'20* | FRCNN-R101 | 25.5 | 20.6 | 45.8 | 14.9 | 32.8 | 29.3 | 22.7 | 27.4 |
| **SED** [40] | *AAAI'21* | FRCNN-V16 | 25.0 | 23.4 | 50.4 | 18.9 | 32.4 | 32.6 | 20.6 | 29.0 |
| **TDD** [23] | *CVPR'22* | FRCNN-V16 | 28.8 | 25.5 | 53.9 | 24.5 | 39.6 | 38.9 | 24.1 | 33.6 |
| **PT** [9] | *ICML'22* | FRCNN-V16 | 28.8 | 33.8 | 52.7 | 23.0 | 40.5 | 39.9 | 25.8 | 34.9 |
| **EPM** [25] | *ECCV'20* | FCOS-R101 | 20.1 | 19.1 | 55.8 | 14.5 | 39.6 | 26.8 | 18.8 | 27.8 |
| **SIGMA** [38] | *CVPR'22* | FCOS-R50 | 26.3 | 23.6 | 64.1 | 17.9 | 46.9 | 29.6 | 20.2 | 32.7 |
| **SIGMA++** [39] | *TPAMI'23* | FRCNN-V16 | 27.1 | 26.3 | 65.6 | 17.8 | 47.5 | 30.4 | 21.1 | 33.7 |
| **NSA** [82] | *ICCV'23* | FRCNN-V16 | / | / | / | / | / | / | / | 35.5 |
| **HT** [14] | *CVPR'23* | FCOS-V16 | 38.0 | 30.6 | 63.5 | 28.2 | 53.4 | 40.4 | 27.4 | 40.2 |
| Baseline | / | FRCNN-R18 | 23.8 | 13.0 | 51.8 | 17.0 | 42.5 | 27.4 | 15.7 | 27.3 |
| **DDT(Ours)** | | | 36.8 | 27.0 | 64.9 | 25.8 | 55.3 | 39.2 | 27.3 | 39.5₊₁₂.₂ |
| Baseline | / | FRCNN-R50 | 24.8 | 16.5 | 53.9 | 15.4 | 45.3 | 27.6 | 18.2 | 28.8 |
| **DDT(Ours)** | | | 39.0 | 31.6 | 65.9 | 30.2 | 57.7 | 39.8 | 28.6 | 41.8₊₁₃.₀ |
| Baseline | / | FRCNN-R101 | 25.9 | 18.4 | 48.8 | 17.2 | 41.1 | 29.8 | 21.7 | 29.0 |
| **DDT(Ours)** | | | 40.3 | 32.3 | 66.7 | 31.8 | 59.1 | 41.6 | 31.8 | **43.4**₊₁₄.₄ |

which we refer to as FRCNN, FCOS, SSD, and DDETR in our table. Furthermore, the backbones with varying depths, including ResNet-18, ResNet-50, ResNet-101 [22], and VGG-16 [60], are denoted as R18, R50, R101, and V16, respectively. To provide a comprehensive comparison, we report the results of our method with ResNet18, ResNet50, and ResNet101. The *baseline* refers to the results that only train on the source domain and test on the target domain.

**Cross-camera adaptation.** Tab. 1 presents the results of the Cross-camera settings. Our DDT method achieved the best performance with mAP 43.4 on the target domain, surpassing the previous SOTA method HT [14] by 3.2 mAP and outperforming other methods by a significant margin. Notably, AT [41] and HT [14] utilize self-training framework, have demonstrated substantial performance improvements by enhancing the quality of generated pseudo labels. Leveraging the powerful feature representation capability of the diffusion model and its exceptional performance on

**Table 2: Quantitative results on adaptation from Sim10K to BDD100K (S→B). The bold indicates the best results.**

| Method | Reference | Detector | mAP(car) |
|---|---|---|---|
| **SWDA** [59] | *CVPR'19* | FRCNN-V16 | 42.9 |
| **CDN** [62] | *ECCV'20* | FRCNN-V16 | 45.3 |
| Baseline | / | FRCNN-R18 | 30.9 |
| **DDT(Ours)** | | | 57.2 +26.3 |
| Baseline | / | FRCNN-R50 | 34.4 |
| **DDT(Ours)** | | | 57.6 +23.2 |
| Baseline | / | FRCNN-R101 | 34.2 |
| **DDT(Ours)** | | | **58.3** +24.1 |

**Table 3: Quantitative results on adaptation from Sim10K to Cityscapes (S→Cs). The bold indicates the best results.**

| Method | Reference | Detector | mAP(car) |
|---|---|---|---|
| **SSAL** [50] | *NeurIPS'22* | FCOS-R50 | 51.8 |
| **O2NET** [19] | *ACMMM'22* | DDETR-R50 | 54.1 |
| **DDF** [44] | *TMM'22* | FRCNN-R50 | 44.3 |
| **D-ADAPT** [29] | *ICLR'22* | FRCNN-R50 | 51.9 |
| **SCAN** [37] | *AAAI'22* | FCOS-V16 | 52.6 |
| **MTTrans** [74] | *ECCV'22* | DDETR-R50 | 57.9 |
| **SIGMA** [38] | *CVPR'22* | FCOS-R50 | 53.7 |
| **TDD** [39] | *CVPR'22* | FRCNN-V16 | 53.4 |
| **MGA** [81] | *CVPR'22* | FCOS-R101 | 54.1 |
| **OADA** [72] | *ECCV'22* | FCOS-V16 | 59.2 |
| **SIGMA++** [39] | *TPAMI'23* | FCOS-V16 | 53.7 |
| **CIGAR** [46] | *CVPR'23* | FCOS-V16 | 58.5 |
| **NSA** [82] | *ICCV'23* | FRCNN-V16 | 56.3 |
| **HT** [14] | *CVPR'23* | FRCNN-V16 | **65.5** |
| Baseline | / | FRCNN-R18 | 42.9 |
| **DDT(Ours)** | | | 62.3 +19.4 |
| Baseline | / | FRCNN-R50 | 43.0 |
| **DDT(Ours)** | | | 62.7 +19.7 |
| Baseline | / | FRCNN-R101 | 43.4 |
| **DDT(Ours)** | | | 64.0 +20.6 |

diverse images, our DDT method achieve a remarkable enhancement in cross-camera detection.

**Synthetic to Real adaptation.** In Tab. 2, our method achieves improvements of 26.3, 23.3, and 24.2 mAP on BDD100K [73] compared with baseline, respectively, surpassing the results of previous algorithms SWDA [59] and CDN [62]. Similarly, our method obtains improvements of 19.4, 19.7, and 19.3 mAP on Cityscapes [12], respectively, surpassing all methods except HT [14] in Table 3. It can be observed that detectors for synthetic-to-real detection, due to the significant differences between synthetic and real-world images, does not perform well with source data only, while our method significantly improves the cross-domain performance from Sim10K [30] to BDD100K [73] and Cityscapes [12].

**Real to Artistic adaptation.** In Tab. 4, 5, and 6, we show the results of real to artistic cross-domain object detection. Our results

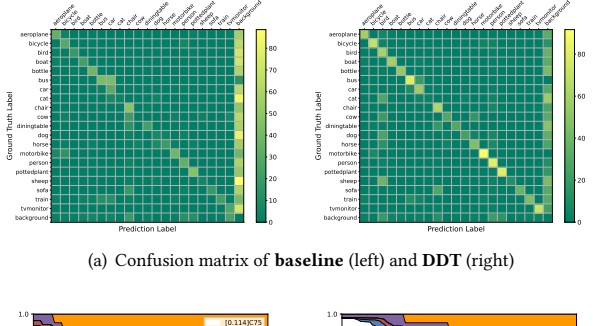

(a) Confusion matrix of **baseline** (left) and **DDT** (right)

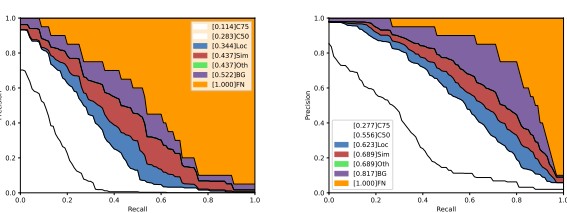

(b) COCO [43] style detection error analysis of the **baseline** (left) and **DDT** (right)

**Figure 4: Error analysis on Clipart.** It is evident that our method significantly reduces false negatives, which correspond to missed detections.

of Resnet50 and ResNet101 significantly surpass the previous best method AT [41] by 3.9 and 4.8 mAP, respectively. On Comic [28], the results of ResNet18, ResNet50, and ResNet101 [22] greatly surpass the previous best result of AT [41], by 2.0, 6.5, and 9.7 mAP, respectively. Similarly, our best result surpasses AT [41] by 3.8 mAP on Watercolor [28] as shown in Tab. 6. In real to artistic benchmark, overall, we find that due to the significant differences between real-world images and artistic-style images, the cross-domain performance is poor. Compared to the baseline, our method exhibits an average relative improvement of 95%, 163%, and 48% on Clipart, Comic, and Watercolor, respectively. This indicates that real to artistic adaptation is a challenging task, and it also demonstrates that our approach we have successfully improved the cross-domain performance by reducing the gap between real and artistic domain.

## 4.5 Ablation Studies

We conduct additional experiments to analyze the feature representation capabilities of different models. Specifically, we compared our diffusion model with powerful backbones, including ConvNext [48], Swin Transformer [47], VIT [15], as well as the self-supervised method MAE [21], pretrained on ImageNet [57]. Additionally, GLIP [33], which is pretrained on a larger dataset and has shown promising performance on object detection benchmarks. Our objective is to investigate two questions:

(1) Will the diffusion model offer better intra-domain and cross-domain feature representation?
(2) Will the diffusion model serve as a better teacher?

**Ablation Study on the Intra-domain and Cross-domain Representation.** To answer the first question, we evaluate the performance of seven models across different data settings in Tab. 7.

**Table 4: Quantitative results on adaptation from VOC to Clipart (V→Ca). The bold indicates the best results.**

| Method | Reference | Detector | aero | bcycle | bird | boat | bottle | bus | car | cat | chair | cow | table | dog | horse | bike | psn | plant | sheep | sofa | train | tv | mAP |
|---|---|---|---|---|---|---|---|---|---|---|---|---|---|---|---|---|---|---|---|---|---|---|---|
| **AT** [41] | CVPR'22 | FRCNN-V16 | 33.8 | 60.9 | 38.6 | 49.4 | 52.4 | 53.9 | 56.7 | 7.5 | 52.8 | 63.5 | 34.0 | 25.0 | 62.2 | 72.1 | 77.2 | 57.7 | 27.2 | 52.0 | 55.7 | 54.1 | 49.3 |
| **D-ADAPT** [29] | ICLR'22 | FRCNN-R50 | 56.4 | 63.2 | 42.3 | 40.9 | 45.3 | 77.0 | 48.7 | 25.4 | 44.3 | 58.4 | 31.4 | 24.5 | 47.1 | 75.3 | 69.3 | 43.5 | 27.9 | 34.1 | 60.7 | 64.0 | 49.0 |
| **TIA** [78] | CVPR'22 | FRCNN-R101 | 42.2 | 66.0 | 36.9 | 37.3 | 43.7 | 71.8 | 49.7 | 18.2 | 44.9 | 58.9 | 18.2 | 29.1 | 40.7 | 87.8 | 67.4 | 49.7 | 27.4 | 27.8 | 57.1 | 50.6 | 46.3 |
| **LODS** [35] | CVPR'22 | FRCNN-R101 | 43.1 | 61.4 | 40.1 | 36.8 | 48.2 | 45.8 | 48.3 | 20.4 | 44.8 | 53.3 | 32.5 | 26.1 | 40.6 | 86.3 | 68.5 | 48.9 | 25.4 | 33.2 | 44.0 | 56.5 | 45.2 |
| **CIGAR** [46] | CVPR'23 | FCOS-R101 | 35.2 | 55.0 | 39.2 | 30.7 | 60.1 | 58.1 | 46.9 | 31.8 | 47.0 | 61.0 | 21.8 | 26.7 | 44.6 | 52.4 | 68.5 | 54.4 | 31.3 | 38.8 | 56.5 | 63.5 | 46.2 |
| **CMT** [3] | CVPR'23 | FRCNN-V16 | 39.8 | 56.3 | 38.7 | 39.7 | 60.4 | 35.0 | 56.0 | 7.1 | 60.1 | 60.4 | 35.8 | 28.1 | 67.8 | 84.5 | 80.1 | 55.5 | 20.3 | 32.8 | 42.3 | 38.2 | 47.0 |
| Baseline | / | FRCNN-R18 | 25.0 | 36.4 | 16.2 | 19.6 | 34.3 | 50.7 | 30.3 | 0.2 | 33.6 | 5.5 | 22.1 | 6.5 | 23.3 | 47.9 | 36.1 | 26.8 | 3.2 | 18.5 | 31.7 | 25.1 | 24.6 |
| **DDT(Ours)** | | | 55.8 | 64.8 | 37.0 | 37.3 | 46.5 | 61.7 | 52.4 | 3.3 | 50.8 | 39.9 | 36.6 | 25.1 | 46.9 | 87.4 | 77.6 | 52.7 | 25.2 | 40.1 | 45.0 | 44.9 | **46.5**+21.9 |
| Baseline | / | FRCNN-R50 | 29.5 | 36.1 | 22.8 | 25.6 | 34.2 | 41.3 | 27.6 | 3.4 | 39.2 | 8.3 | 28.0 | 5.2 | 18.8 | 47.9 | 34.1 | 37.6 | 4.0 | 21.8 | 31.8 | 24.0 | 26.0 |
| **DDT(Ours)** | | | 48.9 | 64.9 | 41.1 | 48.1 | 60.5 | 76.6 | 60.2 | 7.9 | 57.4 | 51.5 | 40.8 | 33.0 | 55.4 | 98.0 | 82.8 | 60.1 | 35.2 | 36.0 | 50.8 | 55.8 | **53.2**+27.2 |
| Baseline | / | FRCNN-R101 | 36.7 | 27.2 | 22.6 | 22.8 | 38.5 | 46.4 | 32.2 | 10.7 | 40.7 | 7.2 | 27.2 | 7.6 | 28.2 | 56.1 | 38.2 | 38.2 | 9.2 | 29.3 | 25.9 | 21.6 | 28.3 |
| **DDT(Ours)** | | | 56.1 | 66.6 | 39.4 | 55.2 | 51.3 | 79.9 | 62.1 | 8.3 | 57.9 | 46.3 | 40.5 | 39.3 | 51.4 | 96.3 | 84.5 | 60.9 | 28.0 | 42.7 | 56.7 | 59.2 | **55.6**+27.3 |

**Table 5: Quantitative results on adaptation from VOC to Comic (V→Co). The bold indicates the best results.**

| Method | Reference | Detector | bicycle | bird | car | cat | dog | person | mAP |
|---|---|---|---|---|---|---|---|---|---|
| **DA-Faster** [10] | CVPR'18 | FRCNN-V16 | 31.1 | 10.3 | 15.5 | 12.4 | 19.3 | 39.0 | 21.2 |
| **SWDA** [59] | CVPR'19 | FRCNN-V16 | 36.4 | 21.8 | 29.8 | 15.1 | 23.5 | 49.6 | 29.4 |
| **STABR** [31] | CVPR'19 | SSD-V16 | 50.6 | 13.6 | 31.0 | 7.5 | 16.4 | 41.4 | 26.8 |
| **MCRA** [79] | ECCV'20 | FRCNN-V16 | 47.9 | 20.5 | 37.4 | 20.6 | 24.5 | 50.2 | 33.5 |
| **I3Net** [7] | CVPR'21 | SSD-V16 | 47.5 | 19.9 | 33.2 | 11.4 | 19.4 | 49.1 | 30.1 |
| **DBGL** [5] | ICCV'21 | FRCNN-R101 | 35.6 | 20.3 | 33.9 | 16.4 | 26.6 | 45.3 | 29.7 |
| **D-ADAPT** [29] | ICLR'22 | FRCNN-R101 | 52.4 | 25.4 | 42.3 | 43.7 | 25.7 | 53.5 | 40.5 |
| Baseline | / | FRCNN-R18 | 29.4 | 6.8 | 11.6 | 5.6 | 7.4 | 26.3 | 14.5 |
| **DDT(Ours)** | | | 56.8 | 21.2 | 46.6 | 25.5 | 32.3 | 72.6 | **42.5**+28.0 |
| Baseline | / | FRCNN-R50 | 37.1 | 6.9 | 29.9 | 6.9 | 10.5 | 30.0 | 20.2 |
| **DDT(Ours)** | | | 60.9 | 28.2 | 52.5 | 29.3 | 36.8 | 74.5 | **47.0**+26.8 |
| Baseline | / | FRCNN-R101 | 37.0 | 6.8 | 31.2 | 4.8 | 7.2 | 26.8 | 19.0 |
| **DDT(Ours)** | | | 63.2 | 34.8 | 56.6 | 31.7 | 39.0 | 75.9 | **50.2**+31.2 |

**Table 6: Quantitative results on adaptation from VOC to Watercolor (V→W). The bold indicates the best results.**

| Method | Reference | Detector | bicycle | bird | car | cat | dog | person | mAP |
|---|---|---|---|---|---|---|---|---|---|
| **SWDA** [10] | CVPR'19 | FRCNN-V16 | 82.3 | 55.9 | 46.5 | 32.7 | 35.5 | 66.7 | 53.3 |
| **MCRA** [80] | ECCV'20 | FRCNN-V16 | 87.9 | 52.1 | 51.8 | 41.6 | 33.8 | 68.8 | 56.0 |
| **UMT** [13] | CVPR'21 | FRCNN-R101 | 88.2 | 55.3 | 51.7 | 39.8 | 43.6 | 69.9 | 58.1 |
| **IIOD** [68] | TPAMI'21 | FRCNN-V16 | 95.8 | 54.3 | 48.3 | 42.4 | 35.1 | 65.8 | 56.9 |
| **I3Net** [7] | CVPR'21 | SSD-V16 | 81.1 | 49.3 | 46.2 | 35.0 | 31.9 | 65.7 | 51.5 |
| **SADA** [11] | IJCV'21 | FRCNN-R50 | 82.9 | 54.6 | 52.3 | 40.5 | 37.7 | 68.2 | 56.0 |
| **CDG** [34] | AAAI'21 | FRCNN-V16 | 97.7 | 53.1 | 52.1 | 47.3 | 38.7 | 68.9 | 59.7 |
| **VDD** [69] | ICCV'21 | FRCNN-V16 | 90.0 | 56.6 | 49.2 | 39.5 | 38.8 | 65.3 | 56.6 |
| **DBGL** [5] | ICCV'21 | FRCNN-R101 | 83.1 | 49.3 | 50.6 | 39.8 | 38.7 | 61.3 | 53.8 |
| **AT** [41] | CVPR'22 | FRCNN-V16 | 93.6 | 56.1 | 58.9 | 37.3 | 39.6 | 73.8 | 59.9 |
| **LODS** [35] | CVPR'22 | FRCNN-R101 | 95.2 | 53.1 | 46.9 | 37.2 | 47.6 | 69.3 | 58.2 |
| Baseline | / | FRCNN-R18 | 71.4 | 36.4 | 39.1 | 19.9 | 12.7 | 52.0 | 38.6 |
| **DDT(Ours)** | | | 81.9 | 53.7 | 54.3 | 37.7 | 31.5 | 68.3 | **54.6**+16.0 |
| Baseline | / | FRCNN-R50 | 68.2 | 40.2 | 44.7 | 21.7 | 10.3 | 44.0 | 38.2 |
| **DDT(Ours)** | | | 96.4 | 58.6 | 52.6 | 33.7 | 36.2 | 71.9 | **58.2**+20.0 |
| Baseline | / | FRCNN-R101 | 72.5 | 40.1 | 45.7 | 30.5 | 18.1 | 45.8 | 42.1 |
| **DDT(Ours)** | | | 87.1 | 64.0 | 55.7 | 50.6 | 48.8 | 75.7 | **63.7**+21.6 |

**Table 7: Results of our diffusion feature extractor (Diff.) compared to other backbones. The bold and underlined represent the best and second performances, respectively**

| Backbones | V→V | Ca→Ca | V→Ca | Rel.(%) | S→S | Cs→Cs | S→Cs | Rel.(%) | B→B | S→B | Rel.(%) |
|---|---|---|---|---|---|---|---|---|---|---|---|
| R101 [22] | 84.0 | 40.0 | 28.3 | 70.8 | 83.0 | 72.6 | 43.4 | 59.8 | 74.2 | 34.2 | 46.1 |
| ConvNext-Base [48] | **91.5** | **62.8** | _44.1_ | 70.2 | **87.9** | **78.2** | **61.8** | _79.0_ | **80.5** | 44.8 | 55.6 |
| Swin-Base [47] | 86.9 | 51.6 | 32.2 | 62.4 | _87.6_ | 73.7 | 53.7 | 72.8 | 79.7 | 39.3 | 49.4 |
| VIT-Base [15] | 84.9 | 31.5 | 28.6 | **90.8** | 77.6 | 72.1 | 48.4 | 67.1 | 73.4 | 40.9 | 55.7 |
| MAE (VIT-Base) [15, 21] | 85.8 | 37.8 | 26.4 | 69.8 | 85.0 | _77.7_ | 57.9 | 74.5 | 78.3 | 40.9 | 52.2 |
| GLIP (Swin-Tiny) [33, 47] | _88.1_ | _55.6_ | 39.9 | 71.6 | 86.5 | 77.2 | _59.3_ | 76.8 | _79.7_ | **50.4** | _63.3_ |
| Diff (Ours) | 75.4 | 43.9 | **47.4** | **108.1** | 76.1 | 71.7 | 58.2 | **81.2** | 71.8 | _50.1_ | **69.8** |

**Table 8: Results of using our diffusion backbone (Diff.) as the teacher model to train student models on different backbones. The bold indicates the best results.**

| Detector Setting | | Cross Domain Settings | | |
|---|---|---|---|---|
| Student | Teacher | V→Ca | S→Cs | S→B |
| R101 [22] | ConvNext-Base [48] | 43.2+19.9 | 60.2+16.8 | 54.6+20.4 |
| R101 [22] | Swin-Base [47] | 35.1+6.8 | 57.5+14.1 | 52.9+18.7 |
| R101 [22] | VIT-Base [15] | 37.4+9.1 | 54.8+11.4 | 51.1+16.9 |
| R101 [22] | MAE (VIT-Base) [15, 21] | 35.8+7.5 | 59.0+16.2 | 53.2+19.0 |
| R101 [22] | GLIP (Swin-Tiny) [33, 47] | 39.6+11.3 | 58.9+15.5 | 54.0+19.8 |
| R101 [22] | Diff. (Ours) | **55.6**+27.3 | **64.0**+20.6 | **58.3**+24.1 |
| ConvNext-Base [48] | Diff. (Ours) | **59.5**+18.4 | 63.5+1.7 | **59.6**+7.1 |
| Swin-Base [47] | Diff. (Ours) | 46.6+14.4 | 63.6+9.9 | 58.6+13.8 |
| VIT-Base [15] | Diff. (Ours) | 41.7+13.1 | 60.2+11.8 | 54.9+15.6 |
| MAE (VIT-Base) [15, 21] | Diff. (Ours) | 43.1+16.7 | **64.6**+6.7 | 57.3+16.4 |
| GLIP (Swin-Tiny) [33, 47] | Diff. (Ours) | 49.5+9.7 | 64.0+4.7 | 59.5+9.4 |

**Table 9: Results of ablation experiments on Diffusion Teacher and Mean Teacher.**

| Settings of Self-training | Cs→B | S→Cs | S→B | V→Ca | V→Co | V→W |
|---|---|---|---|---|---|---|
| DDT (R101) | 43.4 | 64.0 | 58.3 | 55.6 | 50.2 | 63.7 |
| **w/o** Mean Teacher | 40.6-2.8 | 62.5-1.5 | 55.5-2.8 | 49.1-6.5 | 46.2-4.0 | 61.0-2.7 |
| **w/o** Diffusion Teacher | 40.1-3.3 | 57.4-6.6 | 54.4-3.9 | 46.9-8.7 | 37.5-12.7 | 57.4-6.3 |
| **w/o** All Teacher | 37.5-5.9 | 56.1-7.9 | 53.7-4.6 | 41.9-13.7 | 37.4-12.8 | 57.1-6.6 |

Boyong He, Yuxiang Ji, Zhuoyue Tan, and Liaoni Wu.

**Table 10: Ablation results of the diffusion backbone under different time steps and save steps.**

| Time Steps | Save Steps | Train Time (s/iter) | Inf. Time (ms/image) | Cs→B | S→Cs | S→B | V→Ca | V→Co | V→W |
|---|---|---|---|---|---|---|---|---|---|
| 1 | 1 | 0.82 | 271.1 | 29.8 | 57.2 | 50.4 | 37.8 | 36.9 | 52.7 |
| 2 | 2 | 1.14 | 402.5 | 31.0 | 57.5 | 50.1 | 39.3 | 36.3 | 51.8 |
| **5** | **5** | 1.56 | 780.4 | **32.7** | **58.2** | 50.1 | 47.4 | 39.4 | 53.8 |
| 10 | 5 | 2.84 | 1424.2 | 29.8 | 56.3 | **51.1** | 48.4 | 40.9 | 53.7 |
| 20 | 10 | 5.48 | 2710.2 | 28.0 | 55.2 | 50.2 | **48.8** | **41.3** | **54.6** |

Specifically, we compared their intra-domain performance in training and testing within the source domain (**V→V**, **S→S**) and target domain (**Ca→Ca**, **Cs→Cs**, **B→B**), and cross-domain testing (**V→Ca**, **S→Cs**, **S→B**) to assess their cross-domain feature representation capabilities. we find that our diffusion model performe poorly within the intra-domain, lagging behind the other six models. In cross-domain testing, our diffusion model outperformed other methods on Clipart [28] but remained inferior to ConvNext [48] and GLIP [33]. Additionally, we calculate the cross-domain metrics for each model and the results obtained from training and testing in the target domain to measure the relative cross-domain capabilities of each models, represented as "Rel." in Tab. 7. We find that the diffusion model consistently achieves the best relative cross-domain performance. Overall, the answer to the first question may be disappointing, as the diffusion detector showes some improvement in cross-domain performance but still fall behind the detectors with stronger and large-dataset pretrained backbones.

**Ablation Study of Different Teachers.** In Tab. 8, we present the performance of different teacher and student settings for cross-domain detection. First, we use ResNet101 [22] as the student and other models as teacher. We find that although the diffusion model performs worse than ConvNext [48] and GLIP [33] in cross-domain performance, it exhibits significantly better performance when used as a teacher model. Furthermore, when we use the diffusion model as the teacher and the other six models as students, it consistently brings large improvements. This answers our second question, confirming that our diffusion model is indeed a better teacher, even when faced with highly competent students and consistently improve their performance.

**Ablation Results on Diffusion and Mean Teacher.** To better understand the significance of teachers in our DDT, we present the results of different teacher model settings in Tab. 9. The findings reveal that excluding the Mean Teacher and Diffusion Teacher from our method leads to an average decrease of 3.1 and 6.7 mAP, respectively. When all teachers are removed, the self-training performance experiences an average decline of 8.3 mAP. These results clearly demonstrate that both the diffusion teacher and mean teacher play crucial roles in our DDT and are indispensable for achieving better performance. Fig. 1 provides an intuitive illustration of the impacts of the diffusion teacher and mean teacher in training process.

**Ablation Results of Different Diffusion Settings.** We report the results of the diffusion models with different time steps and save steps settings in Tab. 10. It is observed that in cross-domain detection with a larger domain gap (real to artistic), longer time steps and save steps show better results. We consider a trade-off between accuracy and efficiency and choose time steps 5 and save steps 5 as our default settings.

## 4.6 Analysis

**Analysis of Feature Representation of Diffusion Model.** The results in Tab. 7 and 8 further deepen our understanding of the feature representation of diffusion model and its advantages in cross-domain detection. In our view, the observed results can be attributed as: fully frozen weight and adaptation of the diffusion model with the light structure that aligns with the hierarchical feature outputs, limit its performance within the intro-domain compared to fully trainable models. However, when applied as a teacher, the diffusion model guides the student to achieve superior performance in cross-domain, surpassing even the teacher itself. We think that the improved cross-domain representation ability can be attributed to the inherent characteristics of the diffusion model as well as the advantages gained from supervised learning on the source domain. In contrast, other fully trainable teacher models often concentrate primarily on supervised learning on the source domain, resulting in homogeneous optimization and limited guidance for the student. As a result, it becomes challenging to enhance the performance of the students to the level achieved by the homogeneous teacher. These results provide compelling evidence for the advantages of the diffusion model in addressing cross-domain detection tasks.

**Error analysis.** Error analysis on Clipart [28] reveals that false negatives, i.e., missed detections, are the main factor impacting the performance on the target domain as shown in Fig. 4. Our method significantly reduces the number of missed detections, thereby greatly improving the performance of cross-domain detection. A representation of prediction results and feature visualization further corroborate this conclusion, as depicted in the Fig. 3.

## 5 Conclusion

In this paper, we propose a domain adaptive method based on the diffusion model to address the performance degradation caused by the large gap between the source and target domain. We employ a frozen-weight diffusion model as the backbone and extract intermediate feature in the inversion process for the detection task, which we refer to as the diffusion teacher. Subsequently, we apply diffusion teacher in the self-training framework to generate pseudo labels on the unlabeled target domain, guiding the learning of the student model. Our method significantly improves cross-domain detection performance on six datasets, achieving an average improvement of 21.2% mAP compared to the baseline, surpassing the current SOTA methods by an average of 5.7% mAP, without compromising the inference speed. Furthermore, we validate the consistent performance improvement of our method in more extensive experiments for detectors with more powerful backbones, demonstrating the strong and universality domain adaptive capability of our approach.

## 6 Acknowledgment

The authors would like to thank Xiamen University and Unmanned Aerial Vehicle (UAV) Laboratory for the funding and providing with all the necessary technical support.

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
