# OpenReview forum: "Diffusion Domain Teacher: Diffusion Guided Domain Adaptive Object Detector"
_acmmm.org/ACMMM/2024/Conference — MM2024 Poster_

### Official Review · Reviewer_U5jN · 2024-05-21

**Rating:** 5
**Confidence:** 4

**Summary:**

Object detectors face performance challenges due to domain shifts. To address this, the authors propose Diffusion Domain Teacher (DDT), leveraging diffusion models' generative capabilities. DDT utilizes a frozen diffusion model trained on source data to generate pseudo labels for unlabeled target data, guiding supervised learning of a student model. This approach significantly improves cross-domain detection performance.

**Strengths:**

1) Unique use of diffusion models for cross-domain object detection.
2) The teacher-student framework leveraging diffusion models is conceptually sound.
3) Extensive evaluation on multiple datasets shows significant improvements.
4) Well-written and easy to follow.
5) Applicable to cross-domain detection scenarios with limited labeled target data.

**Limitations:**

1) The paper lacks a clear and explicit articulation of the challenges of introducing Diffusion for the proposed approach.
2) It is unclear whether the tested model refers to the student model alone or includes the mean teacher.
3) The paper has a relatively limited number of mathematical formulations, and the default parameter values in Equations (2) and (3) are not sufficiently justified.
4) The bounding boxes shown in Figure 3 for the BDD100k dataset appear imprecisely aligned, indicating potential issues with detection performance.
5) The paper fails to consider a tf baseline for comparison, which could provide further insights.
6) While mAP is used, additional metrics based on Intersection over Union (IoU) could provide a more comprehensive evaluation.
7) Some tables lack comparisons with methods published in recent years (2023, 2024), limiting the paper's ability to demonstrate state-of-the-art status.

**Suitability:**

3

---

### Official Review · Reviewer_6ytp · 2024-05-23

**Rating:** 3
**Confidence:** 3

**Summary:**

This paper focus on cross-domain object detection. The method tackle this problem with a teacher-student framework and improve the quality of pseudo labels with a diffusion model.

**Strengths:**

1. The motivation of proposed method is clear and straightforward.
2. The author conducted extensive comparison experiments on many datasets, and ahcieved sota performance.
3. Extensive ablation study demonstrate the effectiveness of proposed method.

**Limitations:**

Although the proposed method achieves satisfactory performance on multiple domain adaptation scenarios, the paper not provide  a through analysis about the effectiveness. For instance, why the pre-trained diffusion model is more effective than ResNet, ViT?

**Suitability:**

2

---

### Official Review · Reviewer_xySt · 2024-05-24

**Rating:** 5
**Confidence:** 2

**Summary:**

This paper introduces Diffusion Domain Teacher (DDT) for improving cross-domain object detection. It uses a frozen-weight diffusion model trained on the source domain to generate pseudo labels for the target domain, guiding the student model's learning. DDT significantly enhances performance and demonstrates broad applicability and effectiveness in domain adaptation, consistently improving results even in complex models. The paper is well-organized, with rich experiments and significant performance improvements.

**Strengths:**

1.The paper is well-structured and clearly written, making the methodology and results easy to understand.
2.The paper includes a comprehensive set of experiments, validating the effectiveness of the proposed method across multiple benchmarks.
3.The method shows substantial improvements in cross-domain object detection performance, highlighting its practical value.

**Limitations:**

1. The fonts in the figures are not uniform. In Figure 2, some fonts are too dense, and many figures have fonts that are too small to be easily readable. It is recommended to adjust the figures for better readability and ensure they correspond closely with the text.
2. The tables do not highlight the best results, which would make it easier for readers to quickly identify the top-performing methods.
3. The paper does not discuss the limitations of the proposed method, which would be helpful for understanding potential challenges and areas for improvement.

**Suitability:**

2

---

### Meta-Review · Area_Chair_u3E1 · 2024-06-30

**Recommendation:** Accept (Poster)
**Confidence:** 5

**Metareview:**

This paper received two weak accept and one borderline accept final ratings from the reviewers. AC agrees that this paper benefits from good writing and interesting idea. However, the authors are encouraged to make the necessary changes to the best of their ability.